# The Comprehensive Analysis of m6A-Associated Anoikis Genes in Low-Grade Gliomas

**DOI:** 10.3390/brainsci13091311

**Published:** 2023-09-12

**Authors:** Hui Zheng, Yutong Zhao, Hai Zhou, Yuguang Tang, Zongyi Xie

**Affiliations:** Department of Neurosurgery, The Second Affiliated Hospital of Chongqing Medical University, Chongqing Medical University, Chongqing 404100, China; 2022120424@stu.cqmu.edu.cn (H.Z.); zhao_yutong_neuro@yahoo.com (Y.Z.); 2022120425@stu.cqmu.edu.cn (H.Z.); 2021120411@stu.cqmu.edu.cn (Y.T.)

**Keywords:** N6-methyladenosine, anoikis, low-grade glioma

## Abstract

The relationship between N6-methyladenosine (m6A) regulators and anoikis and their effects on low-grade glioma (LGG) is not clear yet. The TCGA-LGG cohort, mRNAseq 325 dataset, and GSE16011 validation set were separately obtained via the Cancer Genome Atlas (TCGA), Chinese Glioma Genome Altas (CGGA), and Gene Expression Omnibus (GEO) databases. In total, 27 m6A-related genes (m6A-RGs) and 508 anoikis-related genes (ANRGs) were extracted from published articles individually. First, differentially expressed genes (DEGs) between LGG and normal samples were sifted out by differential expression analysis. DEGs were respectively intersected with m6A-RGs and ANRGs to acquire differentially expressed m6A-RGs (DE-m6A-RGs) and differentially expressed ANRGs (DE-ANRGs). A correlation analysis of DE-m6A-RGs and DE-ANRGs was performed to obtain DE-m6A-ANRGs. Next, univariate Cox and least absolute shrinkage and selection operator (LASSO) were performed on DE-m6A-ANRGs to sift out risk model genes, and a risk score was gained according to them. Then, gene set enrichment analysis (GSEA) was implemented based on risk model genes. After that, we constructed an independent prognostic model and performed immune infiltration analysis and drug sensitivity analysis. Finally, an mRNA-miRNA-lncRNA regulatory network was constructed. There were 6901 DEGs between LGG and normal samples. Six DE-m6A-RGs and 214 DE-ANRGs were gained through intersecting DEGs with m6A-RGs and ANRGs, respectively. A total of 149 DE-m6A-ANRGs were derived after correlation analysis. Four genes, namely ANXA5, KIF18A, BRCA1, and HOXA10, composed the risk model, and they were involved in apoptosis, fatty acid metabolism, and glycolysis. The age and risk scores were finally sifted out to construct an independent prognostic model. Activated CD4 T cells, gamma delta T cells, and natural killer T cells had the largest positive correlations with risk model genes, while activated B cells were significantly negatively correlated with KIF18A and BRCA1. AT.9283, EXEL.2280, Gilteritinib, and Pracinostat had the largest correlation (absolute value) with a risk score. Four risk model genes (mRNAs), 12 miRNAs, and 21 lncRNAs formed an mRNA-miRNA-lncRNA network, containing HOXA10-hsa-miR-129-5p-LINC00689 and KIF18A-hsa-miR-221-3p-DANCR. Through bioinformatics, we constructed a prognostic model of m6A-associated anoikis genes in LGG, providing new ideas for research related to the prognosis and treatment of LGG.

## 1. Introduction

Glioma is the most common and fatal malignant tumor. Gliomas account for about 80% of all central nervous system malignancies. The WHO classification of the CNS divides gliomas into I–IV grades, of which grades I and II are low-grade and grades III and IV are high-grade [1]. Although low-grade glioma (LGG) accounts for only 20% of cases, its treatment options are affected by many factors. The guidelines for the diagnosis and treatment of LGG are still mainly based on surgery. Postoperative recurrence risk grouping is based on a variety of factors, and subsequent adjuvant radiotherapy and chemotherapy are performed for patients in the high-risk group [2]. However, the molecular characteristics of gliomas are significantly correlated with prognosis and efficacy. LGG has great heterogeneity, and a single clinical diagnosis and treatment idea may not be suitable for all patients [3]. With the deepening of research, it is necessary to divide LGG into different subgroups and explore the new prognostic markers and their molecular mechanisms in LGG. With these, a more targeted and individualized treatment strategy can be developed.

For many years, the medical consensus has been that cancer is a genetic disease, and almost all cancers are caused by genetic changes [4]. In 1974, it was found that influenza viral mRNA contains internal N6-methyladenosine and 5′-terminal 7-methylguanosine in cap structures [5]. Since then, m6A has been found to play an increasingly important role in cancer. The results of pan-cancer analysis showed that there was a significant correlation between m6A and clinical outcome, new epitope burden, immune infiltration, and stemness of the tumor [6]. Anoikis is apoptosis that is induced by inadequate or inappropriate cell-matrix interactions and is involved in tissue-homeostatic, developmental, and oncogenic processes [7]. About 52–62% of patients have a recurrence within 5 years [8]. Resistance to anoikis has been demonstrated to be an important mechanism in the process of glioma invasion and the high recurrence rate [9]. Therefore, exploring the prognostic model of the m6A-associated anoikis gene in LGG can provide a reference for the prognosis evaluation and targeted therapy of LGG patients, improving their prognosis.

We obtained LGG-related data from TCGA, CGGA, and GEO databases and extracted 27 N6-methyladenosine (m6A)-related genes (m6A-RGs) and 508 anoikis-related genes (ANRGs) from the literature, respectively [10,11,12,13]. Finally, we sifted out the risk model genes by differential expression analysis, univariate Cox hazard analysis, and LASSO analysis. and they were subjected to GSEA. Then, an independent prognostic model was constructed, and immune infiltration analysis and drug sensitivity analysis were also performed. Finally, we constructed an mRNA-miRNA-lncRNA regulatory network. The role of the m6A-related anoikis gene prognostic model in LGG was deeply explored, which had great significance for the prognosis and treatment of LGG.

## 2. Materials and Methods

### 2.1. Acquisition of Data

The mRNA expression profiles and clinical information of the TCGA-LGG cohort (5 normal samples, 168 glioblastoma (GBM) samples, and 525 low-grade gliomas (LGG) samples (506 samples with complete survival information and survival time > 0 days)) were acquired via the cancer genome atlas (TCGA) (https://www.cancer.gov/about-nci/organization/ccg/research/structural-genomics/tcga, accessed on 20 January 2022) database. The mRNAseq 325 dataset (PRJCA001746, HRA000073) (182 LGG samples (172 samples with complete survival information)) was downloaded via Chinese Glioma Genome Altas (CGGA) (http://www.cgga.org.cn/index.jsp, accessed on 29 September 2019). The GSE16011 validation set (8 normal samples and 276 grade gliomas (GG) samples (including 24 LGG samples)) was obtained via the Gene Expression Omnibus (GEO) (https://www.ncbi.nlm.nih.gov/gds/?term=GSE16011, accessed on 26 April 2022) database. 27 N6-methyladenosine (m6A)-related genes (m6A-RGs) and 508 anoikis-related genes (ANRGs) were extracted from 3 [10,11,12] and 1 [13] published articles, separately.

### 2.2. Differential Expression Analysis and Functional Enrichment Analysis

Differentially expressed genes (DEGs) between LGG and normal samples in the TCGA-LGG cohort were acquired by the DESeq2 package (Version 1.26.0). [14] setting *p* < 0.05 and |Log_2_FoldChange| > 0.5. Then, differentially expressed m6A-RGs (DE-m6A-RGs) and differentially expressed ANRGs (DE-ANRGs) were gained by taking the intersection of DEGs and 27 m6A-RGs, DEGs, and 508 ANRGs separately. Moreover, gene ontology (GO) and Kyoto Encyclopedia of Genes and Genomes (KEGG) functional enrichment analyses of DE-m6A-RGs and DE-ANRGs were respectively implemented by the clusterProfiler package (Version 4.0.2). [15] setting *p* < 0.05 and count > 1.

### 2.3. Analysis of Protein-Protein Interaction (PPI)

The correlation analysis between DE-m6A-RGs and DE-ANRGs was implemented to acquire DE-m6A-ANRGs by Spearman correlation analysis setting |Cor| > 0.3 and *p* < 0.05. In order to investigate the interaction among DE-m6A-ANRGs, the STRING (https://string-db.org, accessed on 8 February 2023) website was applied to construct the PPI network of them with a confidence level = 0.4.

### 2.4. Construction of the Risk Model

Univariate Cox analysis was implemented based on DE-m6A-ANRGs to sift out genes that had a survival correlation with LGG patients (*p* < 0.05), and 506 LGG samples in the TCGA-LGG cohort were randomly divided into a training set (355 cases) and an internal validation set (151 cases) at 7:3. Then, in the training set, the least absolute shrinkage and selection operator (LASSO) (famil = binomial, type measure = class) was executed based on the genes gained in the previous step to sift out risk model genes. Furthermore, the risk score of each LGG patient was acquired according to the expression of risk model genes and the risk coefficient (obtained via LASSO), and LGG patients were classified into high/low-risk groups through the median risk score (Risk score = α_1_ × X_1_ + α_2_ × X_2_ + … + α_n_ × X_n_), then, a survival analysis was then conducted for these 2 groups. In order to further assess the validity of the risk model, we plotted the receiver operating characteristic (ROC) curve. Finally, the results were validated in the same way in the TCGA internal validation set and the mRNAseq 325 external validation set.

### 2.5. Analyses of Risk Model Genes

First, survival analysis of risk model genes was conducted in the TCGA-LGG cohort, and Kaplan–Meier (K–M) curves were plotted. After that, the positions of risk model genes on chromosomes were analyzed. The risk model genes were treated as target genes, and the correlation coefficients between the expression of all genes in the training set and target genes were calculated and ranked. Gene set enrichment analysis (GSEA) of risk model genes was conducted by clusterProfiler package (Version 4.0.2) [15] setting adj. *p* < 0.05 (50 human hallmarks were selected as gene sets).

### 2.6. Clinical Correlation Analysis

In order to assess the relationship between risk score and risk model genes, we plotted correlation scatter plots. Next, the differences in risk scores in different clinical characteristics groups (age (Years >60 and ≤60), asthma history (NO, YES), eczema history (NO, YES), grade (G2, G3), seizure history (NO, YES), sex (female, male), OS status (0, 1)) were compared separately. At last, we performed a stratified survival analysis on the basis of clinical characteristics.

### 2.7. Construction of the Independent Prognostic Model

Independent prognostic factors were sifted out by including clinical characteristics (gender, age, grade, seizure history, risk score) and risk score into univariate and multivariate Cox. Next, the independent prognostic model was constructed via the cph function, and the nomogram, calibration curve, and ROC curve of it were plotted. Moreover, in order to analyze the effect of high/low-risk groups on LGG progression, we downloaded 50 human gene sets via MSigDB of the Broad Institute (https://www.gsea-msigdb.org/gsea/msigdb/index.jsp, accessed on 8 February 2023), and gene set variation analysis (GSVA) of all genes in these 2 groups was implemented by GSEA (version 3.0.3) software.

### 2.8. Immune Infiltration Analysis and Drug Susceptibility Analysis

In this study, we utilized the ESTIMATE algorithm to infer the proportion of stromal and immune cells in LGG samples of the training set based on gene expression, and the stromal score, immune score, and ESTIMATE composite score were gained after. The ssGSEA (28 immune cell types) and CIBERSORT (20 immune cell types) algorithms were utilized to acquire the infiltration proportions of immune cells in samples, and we analyzed their differences between high/low-risk groups to extract significantly differential immune cells. Then, a correlation analysis between risk model genes and significantly differential immune cells was carried out. Additionally, in order to find potential therapeutic agents for LGG, we calculated the 50% inhibitory concentration (IC50) of common chemotherapeutic agents in LGG samples and compared the differences in IC50 of chemotherapeutic agents (N = 56) between high/low-risk groups by rank sum test. Finally, 198 drugs were identified via genomics of drug sensitivity in cancer (GDSC), and their IC50 in each LGG sample was predicted. Then, we analyzed the correlation between drug IC50 and risk score by Spearman correlation analysis.

### 2.9. Construction of Competing Endogenous RNA (ceRNA) Network

In the TCGA-LGG cohort, differentially expressed miRNAs (DE-miRNAs) and differentially expressed lncRNAs (DE-lncRNAs) between LGG (N = 525) and normal samples (N = 5) were screened out, respectively, by the DESeq2 package (Version 1.26.0) [14] setting *p* < 0.05 and |Log_2_FC| > 1. Secondly, the Starbase website (http://starbase.sysu.edu.cn/, accessed on 8 February 2023) was utilized to predict the miRNAs of risk model genes, and we took the intersection of down-regulated DE-miRNAs and miRNAs to acquire intersected miRNAs. Then, lncRNAs of intersected miRNAs were predicted via the Starbase website, and intersected lncRNAs were gained by taking the intersection of up-regulated DE-lncRNAs and lncRNAs. Finally, we constructed an mRNA-miRNA-lncRNA regulatory network.

### 2.10. External Validation of Risk Model Genes

In order to further validate the expression of risk model genes between LGG and normal samples, expression analysis of them in the TCGA dataset was carried out, and results were validated in the GSE16011 validation set. Finally, the protein expression levels of risk model genes between glioma and normal tissues were analyzed in the public human protein atlas (HPA) database (https://www.proteinatlas.org/, accessed on 10 February 2023).

## 3. Results

### 3.1. Identification and Functional Annotation Analysis of DE-m6A-RGs

There were 6901 DEGs between LGG and normal samples (Figure 1a,b, Appendix A). The DEGs and 27 m6A-RGs were intersected to yield six DE-m6A-RGs, such as ALKBH5 and HNRNPC (Figure 1c). The intersection of DEGs and 508 ANRGs was taken to acquire 214 DE-ANRGs (Figure 1d). DE-m6A-RGs were enriched in 67 GO entries, including regulation of mRNA stability, nucleobase-containing compound catabolic processes, and ribonucleoprotein complex biogenesis (Figure 1e, Appendix A). DE-ANRGs participated in 2363 GO entries, such as response to reactive oxygen species, regulation of nervous system development, peripheral nervous system development, cellular response to nerve growth factor stimulus, and response to nerve growth factor (Figure 1f, Appendix A). DE-ANRGs were engaged in 151 KEGG pathways, including the neurotrophin signaling pathway, glioma, T cell receptor signaling pathway, cellular senescence, and choline metabolism in cancer (Figure 1g, Appendix A).

### 3.2. Characterization of DE-m6A-ANRGs and Construction of the PPI Network

A total of 149 DE-m6A-ANRGs were derived after correlation analysis, such as METTL3, HNRNPC, and ZCCHC4 (Figure 2). Altogether, 149 nodes and 1447 edges formed the PPI network, including LGALS8-ITGA4, EFHD2-FBLIM1, CXCL14-UCHL1, and other interaction pairs (Appendix A). Appendix A illustrates the DE-m6A-ANRGs with a top-20 interaction degree.

### 3.3. Acquisition and Assessment of the Risk Model

A total of 98 genes were identified after univariate analysis, and the top 10 of them are displayed in Figure 3a. When lambda min = 0.12, the error of cross-validation was the lowest, and we acquired 4 risk model genes (ANXA5, KIF18A, BRCA1, and HOXA100) at this point (Figure 3b). It could be seen in the survival curve that LGG patients in the high-risk group had a worse survival (Appendix A and Figure 3c). The area under the curve (AUC) of 1 (AUC = 0.839), 3 (AUC = 0.846), and 5 (AUC = 0.792) years were all greater than 0.7, indicating that the risk model had good predictive efficacy (Figure 3d). The results of the TCGA internal validation set and the mRNAseq 325 external validation set were consistent with the training set (Appendix A).

### 3.4. Risk Model Genes

As seen from the K–M curves, the prognosis in the high-expression group of all 4 risk model genes was worse (Appendix A). ANXA5, KIF18A, BRCA1, and HOXA10 were distributed on No. 4, 11, 17, and 7 chromosomes, respectively (Appendix A). ANXA5 was involved in apoptosis, IL2 STAT5 signaling, and the inflammatory response (Figure 4a, Appendix A). KIF18A is enriched in oxidative phosphorylation, fatty acid metabolism, and E2F target pathways (Figure 4b, Appendix A). BRCA1 was associated with epithelial-mesenchymal transition, mitotic spindle, glycolysis, and so on (Figure 4c, Appendix A). HOXA10 was engaged in the G2M checkpoint, IL6 JAK STAT3 signaling, and interferon alpha response (Figure 4d, Appendix A).

### 3.5. Correlation Analysis of Clinical Characteristics

ANXA5, BRCA1, HOXA10, and KIF18A were all positively correlated with the risk score, with ANXA5 having the highest correlation (R = 0.83) (Figure 5a). From the box plot, it could be concluded that the risk score was dramatically different in age, grade, and OS status (Figure 5b). Figure 5c showed that the survival between high/low-risk groups in age (Years > 60 and ≤60), grade (G2, G3), seizure history (NO, YES), and sex (Female, Male) were significantly different.

### 3.6. Acquisition of the Independent Prognostic Model and GSVA

The age and risk score were finally identified as independent prognostic factors via univariate and multivariate Cox (Figure 6a,b). The C-index of the nomogram was 0.66, indicating that the independent prognostic model had a good prediction of survival at 1-, 3-, and 5-year intervals, and the calibration curve further verified the correctness of the model prediction (Figure 6c,d). The AUC of the nomogram at 1- (AUC = 0.884), 3- (AUC = 0.900), and 5- (AUC = 0.794) years were all higher than those of the age and risk scores, illustrating the best prediction of the nomogram (Figure 6e). The high-risk group was enriched in 15 pathways, including coagulation, estrogen response late, interferon-gamma response, and others. The low-risk group was involved in 14 pathways, such as hypoxia, the P53 pathway, and DNA repair (Figure 6f, Appendix A).

### 3.7. Immune Microenvironment Analysis and Prediction of Potential Therapeutic Agents

In the box plot, we could see that the stromal score, immune score, and estimate score were all dramatically higher in the high-risk group than those in the low-risk group (Figure 7a). A total of 25 immune cells, such as activated B cells, activated dendritic cells, and CD56 bright natural killer cells, were significantly different between high/low-risk groups using the ssGSEA algorithm (Figure 7b–d). Most of the significantly differential immune cells were positively correlated with risk model genes, of which activated CD4 T cells, gamma delta T cells, and natural killer T cells had the greatest positive correlation with risk model genes, while activated B cells were notably negatively correlated with KIF18A and BRCA1 (Figure 7e). Besides, eight immune cells, including naive B cells, CD8 T cells, and CD4 naive T cells, showed differences between high/low-risk groups using the CIBERSORT algorithm (Appendix A). Further, a total of 46 chemotherapeutic agents showed significant differences between high/low-risk groups, and the Top 12 significantly different expressed chemotherapeutic agents in high/low-risk groups were exhibited, such as bexarotene, dasatinib, and erlotinib (Appendix A). AT.9283, EXEL.2280, Gilteritinib, and Pracinostat had the largest correlation (absolute value) with the risk score (Appendix A).

### 3.8. mRNA-miRNA-lncRNA Regulatory Network/

We yielded 347 DE-miRNAs, 1807 DE-lncRNAs, 16 intersected miRNAs, and 21 intersected lncRNAs at last (Figure 8a–d). In total, 4 risk model genes (mRNAs), 12 miRNAs, and 21 lncRNAs formed the ceRNA network, which included HOXA10-hsa-miR-129-5p-LINC00689, KIF18A-hsa-miR-221-3p-DANCR, ANXA5-hsa-miR-410-3p-LINC01579, and other relationship pairs (Figure 8e).

### 3.9. Expression of Risk Model Genes

ANXA5, KIF18A, BRCA1, and HOXA10 were all notably different between LGG and normal samples, and they were highly expressed in LGG samples (Figure 9a). The results in the validation set were consistent with the TCGA dataset (Figure 9b). It can be seen in Figure 9c that proteins corresponding to ANXA5, KIF18A, BRCA1, and HOXA10 were all highly expressed in glioma tissues (Appendix A).

## 4. Discussion

LGG accounts for 6.4% of all primary CNS tumors in adults [16]. The median overall survival (OS) for LGG (WHO grade II), undifferentiated glioblastoma (WHO grade III), and glioblastoma multiforme (WHO grade IV) was 78.1-, 37.6- and 14.4 months, respectively [17]. LGG have higher survival rates and a better prognosis than high-grade gliomas (HGG). Many LGGs inevitably progress to high-grade gliomas. Gliomas can hardly be completely cured by surgical resection [18]. Therefore, it is necessary to explore the mechanism of low-grade glioma metastasis, which can early identify the high-risk group and provide targeted therapy. M6A is the most common mRNA post-transcriptional modification in mammals. This dynamic and reversible epigenetic modification affects the fate of RNA molecules and further regulates mammalian cell differentiation, immunity, and apoptosis. Studies have shown that the imbalance of m6A is involved in the development of gliomas [7,19,20]. M6A can regulate cell self-renewal, differentiation, invasion, and apoptosis by regulating gene expression. Meanwhile, m6A modification in LGG is significantly associated with angiogenesis. Each of the 17 m6A regulators and 36 ARGs exhibited a significant self-positive correlation [20], and the researcher found that three m6a-lncRNAs, including ELOA-AS1, HOXB-AS1, and FLG-AS1, were closely related to the prognosis of LGG patients. Cancer cells can resist anoikis in several ways, such as by secreting growth factors, activating pro-survival signaling pathways, and altering integrin expression patterns [21]. Given the widespread post-translational modification of m6A on RNA molecules, m6A affects the metastasis and deterioration of low-grade gliomas by regulating the expression of anoikis-related genes. Likewise, we also found that seven ANRG genes (ANGPTL2, BAG1, CDH2, IFI27, PTK2B, SOD2, and UBE2C) have a significant relationship with the prognosis of LGG patients. However, the relationship between the m6A regulator and anoikis and their effect on LGG prognosis are still not clear.

Before exploring the prognostic role and biological activities of the m6A-related anoikis gene, it is important to accurately identify these potential genes. Various machine learning (ML) algorithms have been used to predict potential m6A modification sites, such as i6mA-Caps [22], m6Aboost (AdaBoost) [23], as well as periodontitis-associated m6A-SNPs [24]. Considering the empirical application of univariate Cox regression and the LASSO algorithm for constructing diagnostic models, in this study, we screened out four m6A-related anoikis genes (ANXA5, KIF18A, BRCA1, and HOXA10) and found that they were all positively correlated with the risk score. The ANXA5 gene encodes a protein called Annexin A5, which is a calcium-binding protein and belongs to the annexin protein family. ANXA5 is one of the most common proteins that is overexpressed in a series of cancers and participates in the multi-step process of tumor development. It has been reported that annexin A5 plays a role in promoting tumorigenesis and metastasis in various cancers such as pancreatic adenocarcinoma, sarcoma, breast cancer, and prostate cancer [25]. At present, there are few reports about ANXA5 in LGG. A study reported that the ANXA5 polymorphism increased the risk of glioma in patients and suggested a poor prognosis [26]. The research reveals that annexin A5 is the transcriptional target of Snail and can activate the PI3K/Akt/NF-κB signaling pathway [27]. KIF18A encodes the protein of Kinesin family member 18A. This protein is involved in the process of cell mitosis and is an important microtubule dynamin. This study found that the expression of KIF18A was related to the occurrence and development of LGG. The expression level of KIF18A in LGG tissues was higher, which was positively correlated with the size of the tumor and the proliferation ability of the tumor cells [28]. Studies have also found that high expression of KIF18A may indicate an increased risk of malignant transformation from low-grade gliomas to high-grade gliomas [28,29]. BRCA1 is a gene associated with breast cancer that is involved in biological processes such as DNA repair and gene expression regulation. There is no definite study of BRCA1 in gliomas, but it has been reported that the mutation of the BRCA1 gene may be related to some types of brain tumors, such as metastatic brain tumors of breast cancer and ependymoma [29]. BRCA1 gene methylation may also play a crucial role in tumor development [30]. The relationship between LGG and BRCA1 still deserves further study. HOXA10 is a member of the human HOX gene family, which is involved in regulating biological processes such as embryonic development and cell differentiation. At present, the research on HOXA10 in gliomas is still limited. Some studies have shown that the high expression level of HOXA10 is related to the malignancy of gliomas, poor prognosis, and tumor immunity [31,32,33]. All in all, past reports on these genes provide strong evidence for our research. It is worth noting that univariate Cox regression and the LASSO algorithm may neglect the potential interactions and nonlinear relationships among variables, and the inability to handle missing data and outliers might impact the accuracy of the models as well. It is, therefore, necessary to combine larger cohorts and more methods in the future to evaluate the accuracy of the diagnostic model more comprehensively and the interactions among these key genes.

In this paper, we complete the pathway enrichment analysis of the above four genes. There are not many studies on the specific mechanisms of the four genes in LGG or the gene functional pathways involved. Among them, ANXA5 was involved in apoptosis, IL2 STAT5 signaling, and so on. The activation of IL2 receptor binding leads to the activation of JAK protein tyrosine kinase, which promotes the phosphorylation and activation of STAT5 and further regulates the transcription and expression of target genes. A literature review reported that glioma is involved in tumor growth and metastasis through the PI3K/Akt/NF-κB signaling pathway [27]. IL2 STAT5 signaling and IL2 activation can lead to the activation and phosphorylation of PI3K, thereby further activating the AKT signaling pathway and promoting cell proliferation and survival. In the pan-cancer analysis from the perspective of pyroptosis, the IL2-STAT5 signaling pathway was significantly associated with the prognosis of LGG [34]. Inhibition of the IL2-STAT5 signaling pathway can inhibit the proliferation and invasion of LGG cells, promote the apoptosis of tumor cells, and block immune escape. The IL2-STAT5 signaling pathway may become an important therapeutic target in low-grade gliomas.

Another gene we are interested in is BRCA1. BRCA1 has received great attention in breast cancer but little research in gliomas. When the BRCA1 gene mutates or loses its function, it will lead to abnormal biological processes such as DNA damage repair and cell cycle, thereby increasing the risk of the occurrence and development of breast cancer [35]. It has been reported that BRCA1-related proteins inhibit the proliferation and migration of glioma cells and the self-renewal of glioma stem cells through the TGF-β/PI3K/AKT/mTOR signaling pathways [36]. BRCA1-related proteins are significantly related to the prognosis of glioma patients, which is consistent with our research. In low-grade gliomas, the mechanism of BRCA1 in epithelial mesenchymal transition, mitotic spindle, and glycolysis also deserves further study.

In this article, we performed immune infiltrating cell analysis. Studies have shown that low-grade gliomas and high-grade gliomas are significantly different [37]. The decrease in glioma purity is significantly related to malignant entities and a worse prognosis [34]. This is consistent with our results that the stromal score, immune score, and estimate score of the high-risk group were significantly higher than those of the low-risk group. We also noticed that activated CD4 T cells, gamma delta T cells, and natural killer T cells were positively correlated with biomarkers, while activated B cells were negatively correlated with KIF18 A and BRCA1. The presence of activated CD4 T cells, γδ T cells, and NKT cells may indicate that the immune system is trying to recognize and attack tumor cells. This means that high-risk LGGs may trigger a stronger immune response because they have a more aggressive nature. The unique features of γδ T cells make them an emerging breakthrough in cancer immunotherapy. These features include their tissue tropism, tumor-targeting activity independent of neoantigen load, and conventional MHC-dependent antigen presentation, as well as their combination of characteristics typical of unconventional T cells and natural killer cells [38]. Researchers indicate that alterations in the ratio and functionality of Vδ1 T cells and Vδ2 T cells may be linked to the pathogenesis of gliomas [39]. In the presence of IL-2, Vδ1 T cells predominate in patients with glioblastoma, with enhanced immunosuppressive function of Vδ1 T cells and reduced cytotoxicity of Vδ2 T cells. On the other hand, a literature review employed single-cell RNA sequencing profiles from murine glioma models to demonstrate a high proportion of CD4(+), CD8(+) T cells, and natural killer cells in LGG samples, whereas this infiltration was absent in HGG samples [40]. Overall, our findings are consistent with previous results, where specific mechanisms need to be further explored. The study also reported that freshly isolated NKT cells from glioma patients are functionally equivalent to cells from healthy controls [41]. This suggests that activating natural killer T cells to dissolve tumor cells in low-grade gliomas is a possible targeted therapy strategy. Activated B cells are significantly negatively correlated with KIF18 A and BRCA1. At present, there are very few studies on this type of cell. Our results show that tumor invasion may be achieved by inhibiting activated B cells in low-grade gliomas, and its related research mechanism remains to be further studied.

However, our study also has certain limitations. Firstly, there is a lack of functional experimental validation. Secondly, our data is static, which limits our ability to fully comprehend the dynamic changes of the relevant biomarkers. Lastly, the insufficient sample size hinders the precise establishment of predictive models using machine learning. Moving forward, we aim to address these limitations by acquiring larger sample datasets and conducting pertinent experiments to gain a more comprehensive and profound understanding of the regulatory roles of the relevant genes.

## 5. Conclusions

In this paper, a prediction model of m6A-related anoikis genes was constructed by mining public databases, and four biomarkers of ANXA5, KIF18A, BRCA1, and HOXA10 were finally determined, showing strong predictive potential of LGG. Although the validation of clinical samples was lacking, we explored the possible mechanisms in low-grade gliomas by constructing a ceRNA network and performing immune infiltration analysis and drug sensitivity analysis. Next, we will continue to focus on the research progress of cell m6A-related anoikis genes and LGG.

## Figures and Tables

**Figure 1 brainsci-13-01311-f001:**
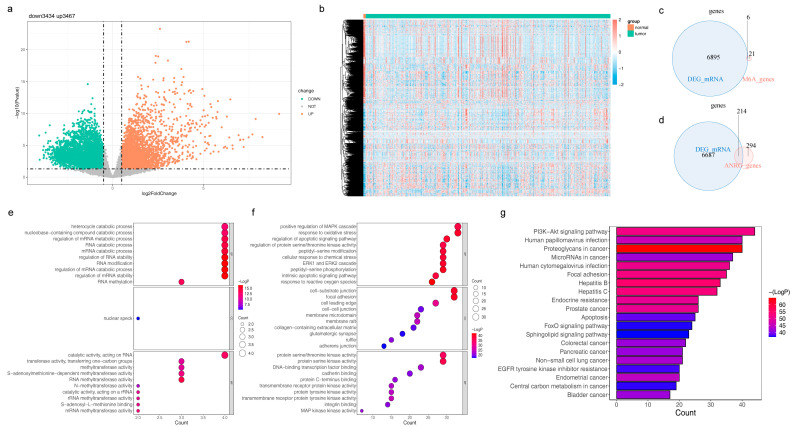
Analysis of six differentially expressed m6A-RGs (DE-m6A-RGs) and 214 differentially expressed anoikis-related genes (DE-ANRGs) in the TCGA-LGG cohort (**a**) Volcano plot and (**b**) heatmap for differentially expressed genes (DEGs) between LGG and normal samples (**c**) Venn plot to identify six DE-m6A-RGs (**d**) Venn plot to identify 214 DE-ANRGs (**e**) Bubble chart for the Gene Ontology (GO) analysis of six DE-m6A-RGs (**f**) Bubble chart for the enriched GO terms of 214 DE-ANRGs (**g**) Bar chart for the Kyoto Encyclopedia of Genes and Genomes (KEGG) analysis of 214 DE-ANRGs.

**Figure 2 brainsci-13-01311-f002:**
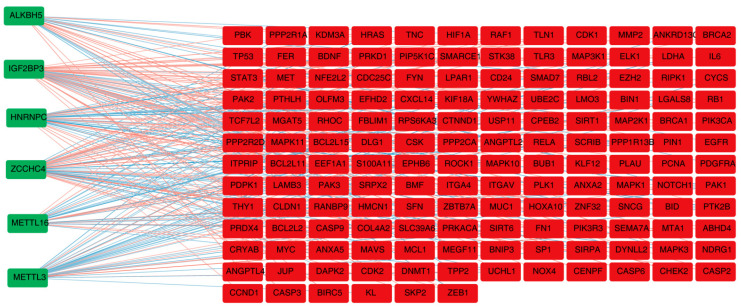
Spearman correlation analysis to identify 149 DE-m6A-ANRGs.Green nodes indicate M6A genes, red nodes indicate M6A-related DE-ANRG. Red edges indicate positive correlation and blue edges indicate negative correlation.

**Figure 3 brainsci-13-01311-f003:**
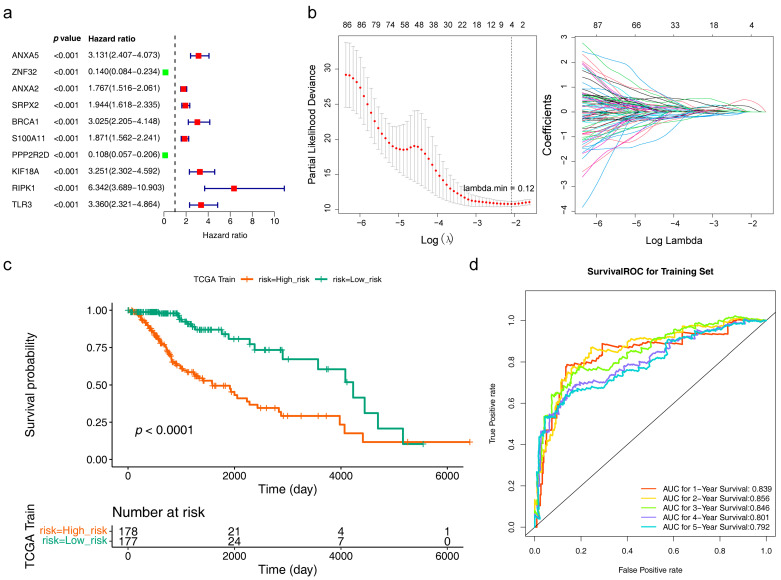
A prognostic model was established based on four risk model genes. (**a**) Univariate Cox regression analysis to screen 10 survival-related genes. Green (HR < 1) indicates the protective factor, red (HR > 1) indicates the risk factor. (**b**) A least absolute shrinkage and selection operator (LASSO) regression model was built based on four risk model genes, including Cross-validation diagram (left) and LASSO coefficients profiles (right). The two vertical dashed lines in the chart are the logλ values corresponding to λmin (the logarithm of the minimum mean square error lambda, the left dashed line) and λ1se (the logarithm of the standard error of the minimum distance lambda, the right dashed line). From left to right along the x-axis, with the increases of lambda, the compression parameter increases and the absolute value of the coefficient decreases. The number on top are the number of variables with non-zero regression coefficients in the LASSO model. Variables with non-zero coefficients are important features for our screening. A line indicates a gene. (**c**) Kaplan–Meier survival curves of the risk model in LGG patients (*p* < 0.0001). Green indicates the low risk groups, red indicates the high risk groups. (**d**) Receiver operating characteristic (ROC) curves for the predictive accuracy of the risk model in LGG patients Different colors indicate the different followed-up years.

**Figure 4 brainsci-13-01311-f004:**
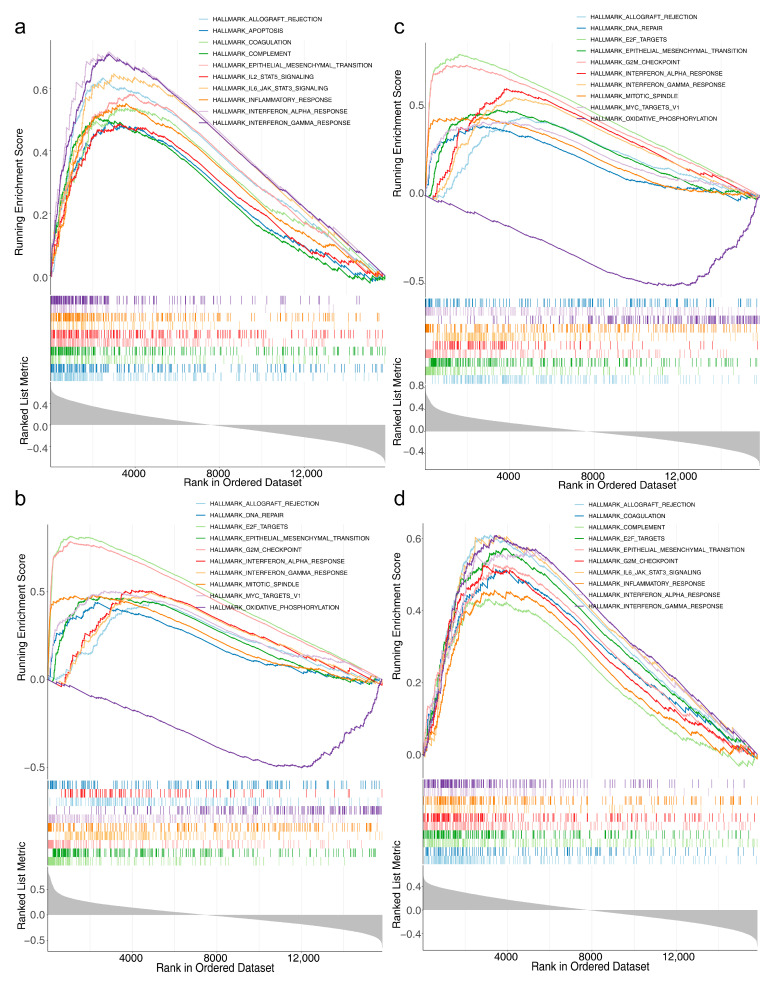
Gene set enrichment analysis (GSEA) of four risk model genes (**a**) Results of GSEA for ANXA5. (**b**) Results of the GSEA of KIF18A (**c**) Results of GSEA for BRCA1. (**d**) Results of GSEA for HOXA10.

**Figure 5 brainsci-13-01311-f005:**
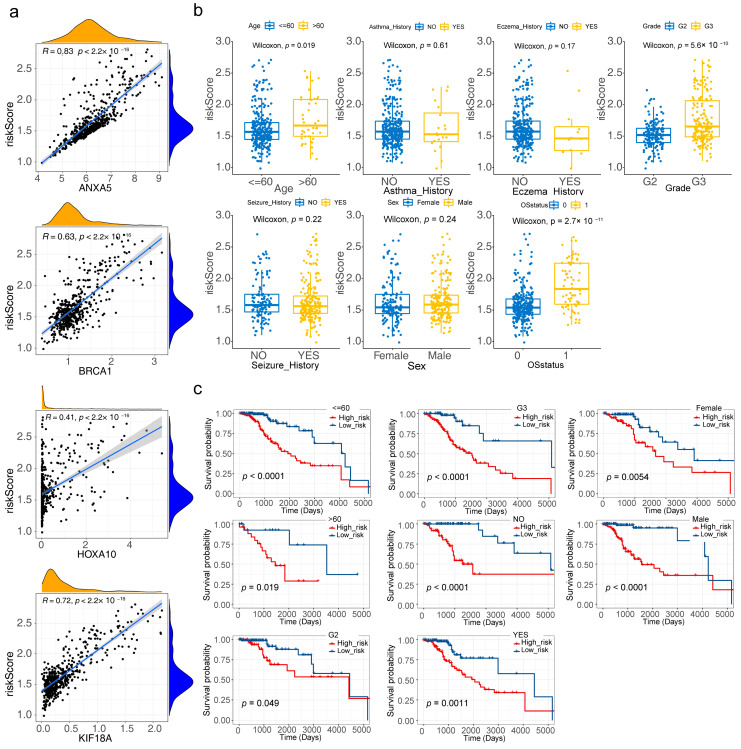
Clinical correlation analysis of four risk model genes and the risk model (**a**) Correlation scatter plots for the relationship between four risk model genes and the risk score (**b**) Boxplot of risk scores in different clinical subtypes (**c**) Clinical stratification analysis for the risk model (*p* < 0.05).

**Figure 6 brainsci-13-01311-f006:**
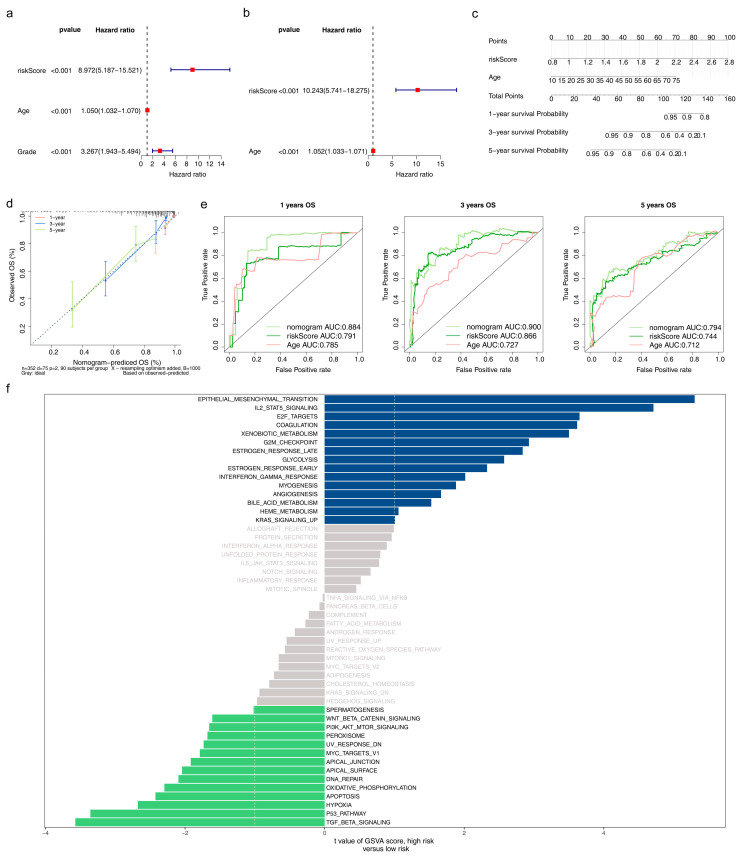
Construction of the nomogram and the effect of high/low-risk groups on LGG progression (**a**) Univariate and (**b**) Multivariate Cox regression analysis to screen independent prognostic factors, including age and risk score. (**c**) The nomogram was built based on the independent prognostic factors. (**d**) Calibration curves of the nomogram to predict survival at 1, 3, and 5 years (**e**) ROC curves to evaluate the predictive accuracy of the nomogram at 1, 3, and 5 years. (**f**) Gene set variation analysis (GSVA) of all genes in the high/low-risk groups.

**Figure 7 brainsci-13-01311-f007:**
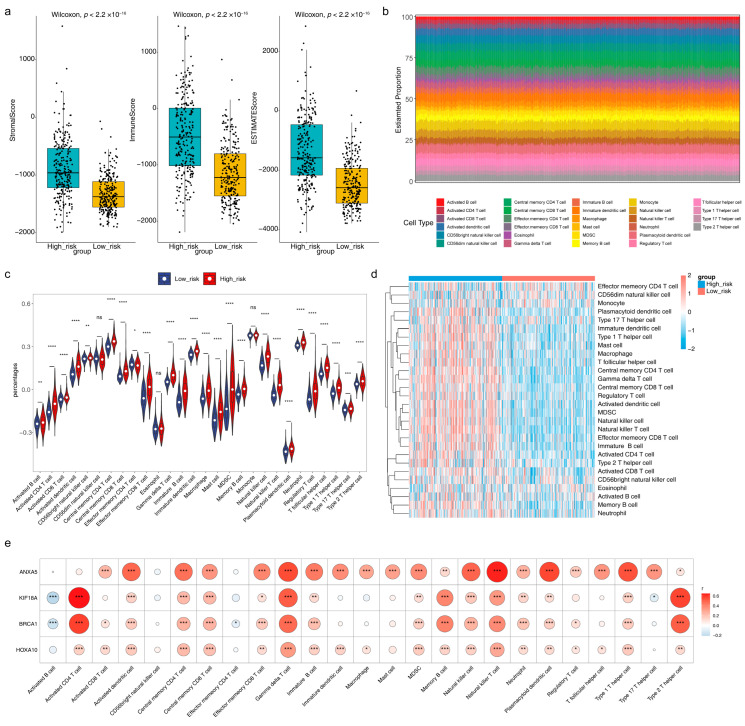
Immune-related analysis and drug susceptibility analysis of the risk model (**a**) Boxplot of the stromal score, immune score, and estimated score in high/low-risk groups (**b**) Histogram for the infiltration score of 28 immune cells in TCGA-LGG cohorts (**c**) Violin plot and (**d**) heatmap for the infiltration levels of 28 immune cells in high/low-risk groups (**e**) Correlation heatmap of four risk model genes and 25 significantly differentially expressed immune cells. ns indicates not significance, * *p* < 0.05, ** *p* < 0.01, *** *p* < 0.001, **** *p* < 0.0001.

**Figure 8 brainsci-13-01311-f008:**
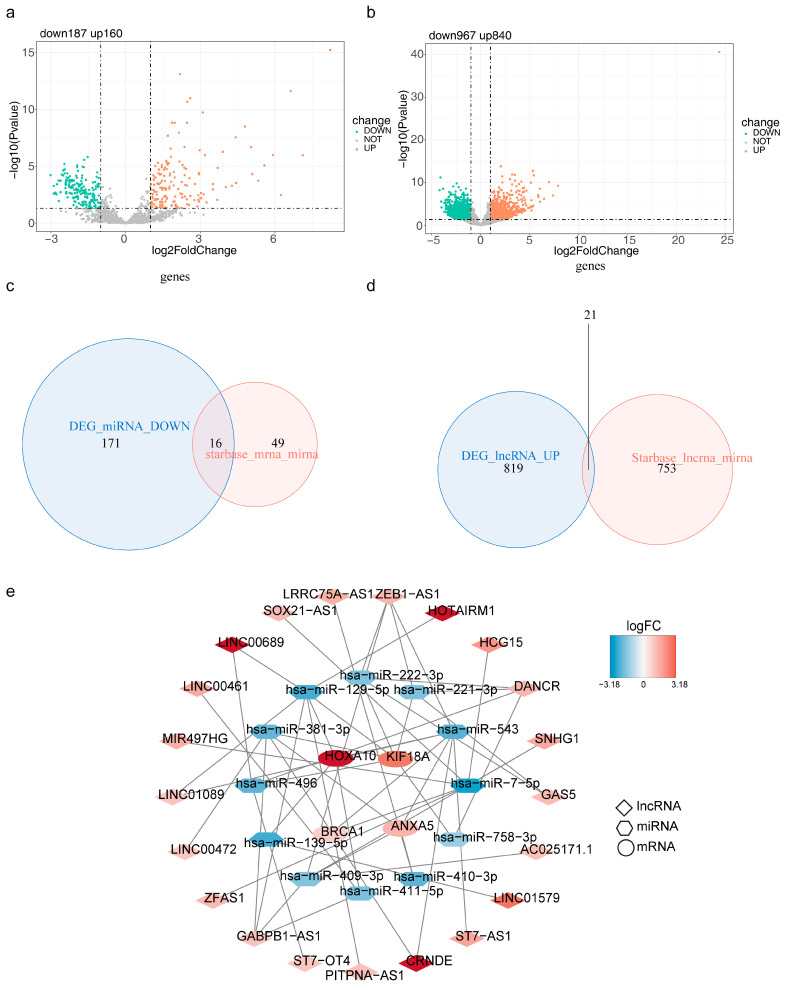
Construction of the mRNA-miRNA-lncRNA regulatory network based on four risk model genes (**a**) Volcano plot for differentially expressed miRNAs (DE-miRNAs) in the TCGA-LGG cohort. (**b**) Volcano plot for differentially expressed lncRNAs (DE-lncRNAs) in the TCGA-LGG cohort. (**c**) Venn plot to identify 16 intersected miRNAs (**d**) Venn plot to identify 21 intersected lncRNAs (**e**) The mRNA-miRNA-lncRNA regulatory network, red represents up-regulated genes, blue represents down-regulated genes.

**Figure 9 brainsci-13-01311-f009:**
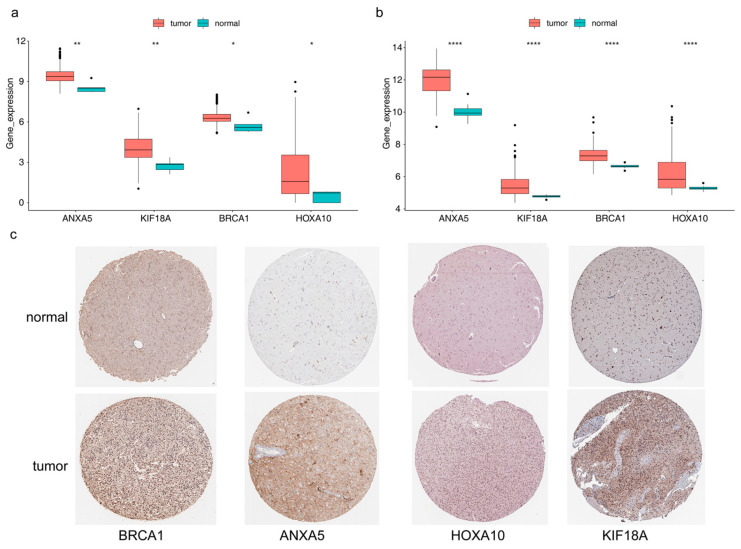
Expression variation of four prognostic genes (**a**) Boxplot of ANXA5, KIF18A, BRCA1, and HOXA10 in the TCGA-LGG cohort (wilcox.test). (**b**) Boxplot of ANXA5, KIF18A, BRCA1, and HOXA10 in the GSE16011 set (wilcox.test), **** *p* < 0.0001, ** *p* < 0.01, * *p* < 0.05. (**c**) The results of immunohistochemistry (IHC) methods for the protein expression levels of risk model genes between glioma and normal tissues through the human protein atlas (HPA) database The deeper the yellow in the diagram, the higher the protein expression of the target gene.

## Data Availability

The datasets generated and/or analyzed during the current study are available in the TCGA (https://www.cancer.gov/about-nci/organization/ccg/research/structural-genomics/tcga, accessed on 20 January 2022), CGGA (http://www.cgga.org.cn/index.jsp, accessed on 29 September 2019) and GEO (https://www.ncbi.nlm.nih.gov/gds, accessed on 26 April 2022) database. The immunohistochemistry (IHC) images for the protein expression levels of risk model genes between Glioma and Normal tissues were obtained from the human protein atlas (HPA) database (https://www.proteinatlas.org/, accessed on 10 February 2023).

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
