# Peer review of "The Comprehensive Analysis of m6A-Associated Anoikis Genes in Low-Grade Gliomas"

_brainsci, 2023, doi:10.3390/brainsci13091311_

Round 1
Reviewer 1 Report
The manuscript is very well written but I believe there are a few points that need to be addressed,
@ How about Including time-series data to capture dynamic changes during pluripotent reprogramming, enhancing the understanding of m6A effector involvement.
@ What do authors think about Utilizing machine learning algorithms to predict potential m6A modification sites, offering insights into novel regulatory elements.
@ Though if authors don't want to add it in the manuscript but I would like them to add a paragraph in the manuscript firstly discussing the ML models being used for identifying methylation and then highlighting the limitation that ML algorithms hold today to solve this problem. The authors should use the following manuscripts in that paragraph,
https://doi.org/10.1093/bioinformatics/btac434
https://doi.org/10.1093/nar/gkab485
https://doi.org/10.1002/jcp.29005
@ How about conducting functional knockdown experiments of m6A effectors to validate their role in pluripotent reprogramming and explore their effects on downstream gene expression.
Its fine
Author Response
Dear Editor and Reviewers,
Thank you very much for giving us opportunities to revise our manuscript, and we appreciate the reviewer a lot for his positive and constructive comments and suggestions. We have studied reviewer’s comments carefully and have made revisions We hope the corrections will meet with your approval.
Question: How about Including time-series data to capture dynamic changes during pluripotent reprogramming, enhancing the understanding of m6A effector involvement.
Response: Thank you for the review. Incorporating time-series data can indeed provide a more comprehensive understanding of the dynamic changes in key m6A-related biomarkers during pluripotent reprogramming, as you mentioned. However, considering the consistency of the data, in this study, we primarily leveraged RNA-seq data from public datasets to select critical genes that may have diagnostic value in low-grade gliomas (LGGs). We employed external expression validation, encompassing mRNA and protein levels, to establish a theoretical foundation for constructing a nomogram model for clinical application based on their expression profiles. Simultaneously, through immune-related analysis, we were able to identify genes associated with immune cell proportions, which can serve as a theoretical basis for subsequent front-end evaluation of gene dynamic expression changes using single-cell analysis profiling. In accordance with your suggestions, we have addressed the limitations of this study and plan to utilize MeRIP-seq or m6A-seq techniques in the future, measuring m6A modifications at different time points, thereby capturing the global dynamic changes of m6A effectors at the transcriptional level through the collection of additional time-series data
Question: What do authors think about Utilizing machine learning algorithms to predict potential m6A modification sites, offering insights into novel regulatory elements.
Response: Thank you for the feedback. The utilization of machine learning algorithms to predict potential m6A modification sites is closely intertwined with the process of machine learning-based selection of pivotal m6A-related diagnostic biomarkers. Executing this analysis necessitates the compilation of extensive datasets comprising sequence information and other pertinent features for both m6A-modified and non-modified sites, including omics data associated with m6A modifications. By employing appropriate feature selection methods, one can filter out features with significant impact on m6A modifications, subsequently determining the most valuable attributes for predicting m6A modification sites.
However, taking data consistency into account, our study primarily seeks to leverage transcriptomic data. Our objective is to employ the Univariate Cox regression-LASSO analysis method to identify key m6A-apoptosis related genes relevant to low-grade glioma (LGG) diagnosis. This endeavor facilitates the construction of an accurate machine learning model that associates critical genes with diagnostic outcomes. In response to your advice, we have addressed the limitations of our study. Furthermore, in the future, we intend to tap into larger sample cohorts to further explore latent m6A modification sites, thereby expanding novel insights into regulatory elements associated with key genes.
Question: Though if authors don't want to add it in the manuscript but I would like them to add a paragraph in the manuscript firstly discussing the ML models being used for identifying methylation and then highlighting the limitation that ML algorithms hold today to solve this problem. The authors should use the following manuscripts in that paragraph,
Response: Thanks for your suggestions. Based on your suggestions, we have updated the Discussion section as follows:
Before exploring the prognostic role and biological activities of m6A-related anoikis gene, it is important to accurately identify these potential genes. Various machine learning (ML) algorithms has been used to predict potential m6A modification sites, such as i6mA-Caps [1], m6Aboost (AdaBoost) [2], as well as periodontitis -associated m6A-SNPs were identified[3] . Considering the empirical application of univariate Cox regression and LASSO algorithm for constructing diagnostic models, in this study, we screened out four m6A related anoikis genes (ANXA5, KIF18A, BRCA1 and HOXA10), and found that they were all positively correlated with the risk score.
It is worth that univariate Cox regression and the LASSO algorithm may neglect the potential interactions and nonlinear relationships among variables, and the incapability to handle missing data and outliers might impact the accuracy of the models as well, it is necessary to combine larger cohorts and more methods in the future to evaluate the accuracy of the diagnostic model more comprehensively and the interactions among these key genes.
Question: How about conducting functional knockdown experiments of m6A effectors to validate their role in pluripotent reprogramming and explore their effects on downstream gene expression.
Response: Thanks for your suggestions. In this study, our primary focus was on bioinformatics analysis, complemented by gene expression validation, including both mRNA and protein levels. Our emphasis lay in examining the expression levels of key m6A effectors within LGG samples. Simultaneously, we integrated GSEA analysis to predict crucial functional pathways associated with these m6A effectors in LGG. In regard to functional experiments, we acknowledge the need for functional knockdown experiments to further explore the impact of key m6A effectors on pluripotent reprogramming and downstream target gene expression. However, considering factors like limited funding, as per your suggestions, we have supplemented the limitations of our study.
- Rehman, M.U., et al., i6mA-Caps: a CapsuleNet-based framework for identifying DNA N6-methyladenine sites. Bioinformatics, 2022. 38(16): p. 3885-3891.
- Körtel, N., et al., Deep and accurate detection of m6A RNA modifications using miCLIP2 and m6Aboost machine learning. Nucleic Acids Res, 2021. 49(16): p. e92.
- Lin, W., et al., In silico genome-wide identification of m6A-associated SNPs as potential functional variants for periodontitis. J Cell Physiol, 2020. 235(2): p. 900-908.
Reviewer 2 Report
Zheng et al have evaluated the relationship between N6-methyadenosine regulators and anoikis in low-grade glioma (LGG). Importantly, they established a prognostic model based on 4 risk model genes. Furthermore, different immune cell types and potential therapeutic targets in high and low-risk groups were analyzed. Overall, this is a very extensive manuscript, and the results are quite interesting. However, please address the following concerns
Major comments
· Table S1 is missing in the additional file.
· The quality of all the figures needs to be improved. The resolution is of low quality to read and review.
· Please provide the list of genes that are downregulated when DEGs are compared between LGG and normal samples. Similarly, also include downregulated m6A-RGs and ANRGs.
· As immune findings are an important aspect of this manuscript. To complement the findings from the ESTIMATE algorithm, can authors perform CIBERSORT or CIBERSORTx analysis to deconvolute the abundance of different immune cell types in LGGs?
· Can authors comment on why the high-risk groups have a positive correlation with activated CD4 T cells, gamma delta T cells, and NKT cells? Overall, from the heatmap shown in Fig 7D, it appears there is increased infiltration of immune cells in the high-risk group compared to the low-risk group. Can the authors comment on why this is the case? These observations contrast with what is described in lines 395-396.
· In Figure 9 legends, please include what statistics were performed to determine significance. How many tissues were analyzed and what tissues are considered normal in Figure 9C? Please also provide quantification for 9C and add details on how IHC was performed in the methods section.
Minor comments
· In line 61, the authors describe that resistance to anoikis as an important mechanism for tumor metastasis. Therefore, the m6A-assosciated anoikis gene in LGG can provide a reference for prognosis evaluation in LGG. Additionally, low-grade glioma metastasis was described in lines 319 and 334. Can authors provide information on how much percent of LGGs metastasize?
Author Response
Dear Editor and Reviewers,
Thank you very much for giving us opportunities to revise our manuscript, and we appreciate the reviewer a lot for his positive and constructive comments and suggestions. We have studied reviewer’s comments carefully and have made revisions We hope the corrections will meet with your approval.
Major comments
Question: Table S1 is missing in the additional file.
Response: Thanks for the comment. We have supplemented the file in additional file. You can verify it yourself
Question: The quality of all the figures needs to be improved. The resolution is of low quality to read and review.
Response: I apologize for our negligence. Based on your suggestions, we have updated the Figure section to increase the resolution to comply with journal requirements
Question: Please provide the list of genes that are downregulated when DEGs are compared between LGG and normal samples. Similarly, also include downregulated m6A-RGs and ANRGs.
Response: Thanks for the comment. Please see the file Downregulated genes.zip for a list of these.
Question: As immune findings are an important aspect of this manuscript. To complement the findings from the ESTIMATE algorithm, can authors perform CIBERSORT or CIBERSORTx analysis to deconvolute the abundance of different immune cell types in LGGs?
Response: Thanks for the comment. Based on your recommendations, we additionally perform CIBERSORT analysis to supplement the abundance of different immune cell types as seen in Figure S4.
Question: Can authors comment on why the high-risk groups have a positive correlation with activated CD4 T cells, gamma delta T cells, and NKT cells? Overall, from the heatmap shown in Fig 7D, it appears there is increased infiltration of immune cells in the high-risk group compared to the low-risk group. Can the authors comment on why this is the case? These observations contrast with what is described in lines 395-396.
Response: I apologize for our negligence. At your suggestion, we first changed the relevant statements in the manuscript as follows:
The decrease of glioma purity is significantly related to malignant entities and worse prognosis[34]. This is consistent with our results that the Stromal score, Immune score and ESTIMATE score of the high-risk group was significantly higher than that of the low-risk group.
It can be seen that the stroma score, immune score and ESTIMATE score of patients in the high-risk group of LGG are high, while the tumor purity is low and the prognosis is poor, which is consistent with the previous research results. Second, based on Figure 7B-D, we found an increased proportion of activated CD4 T cells, γδ T cells, and NKT cells in LGG high-risk groups, possibly to combat tumor cell proliferation and invasion. These immune cells enhance the immune response through various mechanisms, inhibiting tumor growth and spread. At the same time, the high expression trend of key genes in tumors does not contradict the results of their positive correlation with these cells, but the substantial causal relationship needs to be further studied and explored. Based on your suggestions, we have added the relevant statements in the Discussion section as follows:
The presence of activated CD4 T cells, γδ T cells, and NKT cells may indicate that the immune system is trying to recognize and attack tumor cells. This means that high-risk LGGs may trigger a stronger immune response because they have a more aggressive nature. The unique features of γδ T cells make them an emerging breakthrough in cancer immunotherapy. These features include their tissue tropism, tumor-targeting activity independent of neoantigen load and conventional MHC-dependent antigen presentation, as well as their combination of characteristics typical of unconventional T cells and natural killer cells [1].PMID: [2] indicates that alterations in the ratio and functionality of Vδ1 T cells and Vδ2 T cells may be linked to the pathogenesis of gliomas. In the presence of IL-2, Vδ1 T cells predominate in patients with glioblastoma, with enhanced immunosuppressive function of Vδ1 T cells and reduced cytotoxicity of Vδ2 T cells.On the other hand, PMID: [3] employed single-cell RNA sequencing profiles from murine glioma models to demonstrate a high proportion of CD4(+), CD8(+) T cells, and natural killer cells in LGG samples, whereas this infiltration was absent in HGG sample.Overall, our findings are consistent with previous results, where specific mechanisms need to be further explored.
Question: In Figure 9 legends, please include what statistics were performed to determine significance. How many tissues were analyzed and what tissues are considered normal in Figure 9C? Please also provide quantification for 9C and add details on how IHC was performed in the methods section.
Response: Thanks for the comment. Based on your suggestions, we have updated Figure 9A-B to Figure legends as follows:(A) Boxplot of ANXA5, KIF18A, BRCA1 and HOXA10 in TCGA-LGG cohort (wilcox.test). (B) Boxplot of ANXA5, KIF18A, BRCA1 and HOXA10 in GSE16011 set (wilcox.test).In addition, the IHC results in Figure 9C are derived from an online database (HPA database, https://www.proteinatlas.org) that captures protein expression levels of 4 key genes in gliomas and normal brain tissue (with the cerebral cortex as a normal control sample). It is worth noting that the staining results of protein immunohistochemistry in the HPA database are divided into four categories: according to Antibody Staining, it is divided into: High, Medium, Low, Not detected; According to Intensity, it is divided into: Strong, Moderate, Weak, Negative. Therefore, it does not involve calculating detailed quantitative results and disclosing experimental methods. In this study, we selected representative immunohistochemical staining maps of samples from the corresponding staining results of 4 key genes for display. The corresponding antibody information and detailed staining results can be consulted in the HPA online database, and the expression results of the corresponding genes are shown in Table S10.
Minor comments
Question: In line 61, the authors describe that resistance to anoikis as an important mechanism for tumor metastasis. Therefore, the m6A-assosciated anoikis gene in LGG can provide a reference for prognosis evaluation in LGG. Additionally, low-grade glioma metastasis was described in lines 319 and 334. Can authors provide information on how much percent of LGGs metastasize?
Response: Thanks for your comment. What we want to express is that resistance to anoikis as an important mechanism for the diffuseness of glioma invasion and the high recurrence rate. About 52%–62% of patients have a recurrence within 5 years([4]). We have corrected the relevant statements in the text.
- Godfrey, D.I., et al., Unconventional T Cell Targets for Cancer Immunotherapy. Immunity, 2018. 48(3): p. 453-473.
- Yue, C., et al., γδ T Cells in Peripheral Blood of Glioma Patients. Med Sci Monit, 2018. 24: p. 1784-1792.
- Rajendran, S., et al., Single-cell RNA sequencing reveals immunosuppressive myeloid cell diversity during malignant progression in a murine model of glioma. Cell Rep, 2023. 42(3): p. 112197.
- Sanai, N., S. Chang, and M.S. Berger, Low-grade gliomas in adults. J Neurosurg, 2011. 115(5): p. 948-65.
Reviewer 3 Report
The objective of the authors was to investigate the relationship between m6A regulator and anoikis related gene signature and their impact on low grade glioma (LGG). The authors have used publicly available datasets from TCGA, Chinese Glioma Genome Altas (CGGA) and Gene Expression Omnibus (GEO) databases to extract the data on m6A and anikis related gene signature. The authors have analyzed differentially expressed genes and correlation analysis was performed to identify differentially expressed m6A-ANRGs. The authors have performed analysis to identify risk model genes and constructed independent prognostic models. The authors have also performed immune filtration analysis and drug sensitivity analysis. It is interesting study, and the authors are requested to address following concerns.
Major concerns:
1) The major weakness of this study is lack of validation data set nonetheless it is interesting data using bioinformatic analysis.
2) Please include the identifiers for all datasets used by the authors in the study. The authors have cited the papers but the authors should include the identifiers details as well.
Minor concerns:
1) Please fix the font size for text in Ln # 59 - 65
2) The authors are requested to paraphrase sentences and consider avoiding the use of words such as "etc" or "et. al." to describe the data.
The authors are requested to carefully proofread the manuscript for syntax errors.
Author Response
Dear Editor and Reviewers,
Thank you very much for giving us opportunities to revise our manuscript, and we appreciate the reviewer a lot for his positive and constructive comments and suggestions. We have studied reviewer’s comments carefully and have made revisions We hope the corrections will meet with your approval.
Major concerns:
Question: The major weakness of this study is lack of validation data set nonetheless it is interesting data using bioinformatic analysis.
Response: Thank you for your feedback. In this study, we primarily employed TCGA-LGG as the training dataset and validated the prognosis model (mRNAseq 325) as well as the expression levels of prognostic model genes (GSE16011) using mRNAseq 325 and GSE16011 as validation datasets, respectively. Additionally, we utilized the HPA database's immunohistochemistry staining profiles to assess protein expression levels. However, regarding the sample size, we acknowledge your point; further validation of our findings will require the collection of updated and larger sample cohorts. Once again, we appreciate your suggestions.
Question: Please include the identifiers for all datasets used by the authors in the study. The authors have cited the papers but the authors should include the identifiers details as well.
Response: Thank you for your review. As we were not entirely clear about the issues you pointed out, based on your advice, we have provided more specific information about the search link for the GSE16011 dataset: https://www.ncbi.nlm.nih.gov/gds/?term=GSE16011. Additionally, we have included alternative identifiers for the mRNAseq 325 dataset: PRJCA001746, HRA000073, which you can refer to for further details.
Minor concerns:
Question: Please fix the font size for text in Ln # 59 - 65
Response: I apologize for our oversight. Based on your suggestions, we have verified the text formatting in the manuscript and corrected the font size.
Question: The authors are requested to paraphrase sentences and consider avoiding the use of words such as "etc" or "et. al." to describe the data.
Response: Thank you for your feedback. We have addressed the issues in the wording as per your suggestions.
Round 2
Reviewer 2 Report
The authors have addressed all my comments.